# Associations between dietary patterns and intestinal inflammation among HIV-infected and uninfected adults: A cross-sectional study in Tanzania

**Evangelista Kenan Malindisa**[1,2]\*, **Haruna Dika**[1], **Andrea Mary Rehman**[3], **Belinda Kweka**[2], **Jim Todd**[2], **Mette Frahm Olsen**[4,5], **Rikke Krogh-Madsen**[6,7,8], **Ruth Frikke-Schmidt**[6,9], **Henrik Friis**[5], **Daniel Faurholt-Jepsen**[4,6], **Paul Kelly**[10,11], **Suzanne Filteau**[3], **George PrayGod**[2]

1 Department of Physiology, the Catholic University of Health and Allied Sciences, Mwanza, Tanzania, 2 Mwanza Research Centre, National Institute for Medical Research, Mwanza, Tanzania, 3 Faculty of Epidemiology and Population Health, London School of Hygiene & Tropical Medicine, London, United Kingdom, 4 Department of Infectious Diseases, Rigshospitalet, Copenhagen, Denmark, 5 Department of Nutrition, Exercise and Sports, University of Copenhagen, Copenhagen, Denmark, 6 Department of Clinical Medicine, University of Copenhagen, Copenhagen, Denmark, 7 Centre for Physical Activity Research, Rigshospitalet, University of Copenhagen, Denmark, 8 Department of Infectious Diseases, Copenhagen University Hospital, Hvidovre, Copenhagen, Denmark, 9 Department of Clinical Biochemistry, Rigshospitalet, Copenhagen, Denmark, 10 Tropical Gastroenterology and Nutrition group, University of Zambia School of Medicine, Lusaka, Zambia, 11 Blizard Institute, Barts & The London School of Medicine, Queen Mary University of London, London, United Kingdom

\* maryvianey12@gmail.com

**Data Availability Statement:** Data including de-identified data cannot be shared publicly because, according to Tanzanian ethics guidelines, it is not

## Abstract

The increased burden of non-communicable diseases (NCDs) is fueled by lifestyle factors including diet. This cross-sectional study explored among Tanzanian adults whether unhealthy dietary patterns are associated with intestinal and systemic inflammation which could increase the risk of NCDs. The study included 574 participants, with both diet and inflammatory markers data. Dietary patterns were derived using principal component analysis and reduced rank regression, revealing three main patterns: vegetable-rich, vegetable-poor, and carbohydrate-dense diets. Fecal myeloperoxidase (MPO) and neopterin (NEO) were markers of intestinal inflammation whereas plasma lipopolysaccharide-binding protein (LBP) and C-reactive protein (CRP) were assessed as markers of systemic inflammation. Ordinal logistic regression was used to assess associations between terciles of dietary patterns and quintiles of the inflammatory markers adjusting for potential confounders. High adherence to a vegetable-poor dietary pattern was associated with elevated MPO (adjusted OR, 1.7 95% CI 1.1, 2.8). NEO tended to be higher in people with high adherence to both vegetable-poor pattern (adjusted OR, 2.6 95% CI 1.0, 6.4) and vegetable-rich pattern (adjusted OR, 2.7, 95% CI 1.1, 6.5). No associations were found between dietary patterns and systemic inflammation markers (LBP and CRP). We found links between dietary vegetable intake and intestinal inflammation but not systemic inflammation. However, the cross-sectional nature of the study limits establishing causality and the sample size for some

possible to share any data including de-identified data without approval of the Medical Research Coordinating Committee (MRCC). Data are available from the National Institute for Medical Research (NIMR) and can be shared with researchers who meet the criteria for access to confidential data only after completing a data transfer agreement and approval by the MRCC. The MRCC can be contacted at ethics@nimr.or.tz.

**Funding:** This work received fund from the European & Developing Countries Clinical Trials Partnership (EDCTP2) program supported by the European Union (grant agreement number: TMA2017GSF-1965-REEHAD), awarded to GP. It received additional support from the Ministry of Foreign Affairs of Denmark through the DANIDA Fellowship Centre (grant: 16-P01-TAN) awarded to GP. The funders are not responsible for any use that may be made of the information contained in this publication, the funders had no role in the study design, data collection and analysis, decision to publish, or preparation of the manuscript.

**Competing interests:** The authors have declared that no competing interests exist.

variables may have been inadequate, emphasizing the need for further studies to understand how dietary habits influence inflammation in this population.

## Introduction

The burden of non-communicable diseases, including diabetes, is increasing worldwide, especially in low- and middle-income countries [1]. Lifestyle factors such as intake of energy-dense foods and physical inactivity are thought to be major contributors to the increasing diabetes burden [2] although studies are scarce in sub-Saharan Africa (SSA) [3]. Investigating dietary patterns offers a more holistic view of dietary habits compared to examining individual foods or nutrients [4]. Dietary patterns consider the combined effect of multiple dietary components and their interactions, which may have synergistic or antagonistic effects on health [5]. This approach better reflects real-world eating behaviors and can provide insights into the complex relationships between diet and health outcomes, such as intestinal and systemic inflammation [6].

We recently found that among adults in Tanzania, a carbohydrate-dense diet was associated with an increased risk of insulin resistance, prediabetes, and diabetes, but not beta-cell dysfunction [7]. The association could not have been mediated by obesity [8, 9] since both low and high body mass index (BMI) were associated with prediabetes and diabetes [10], indicating that there may be other causal pathways between diet and diabetes [11, 12]. Understanding mechanisms linking diet and diabetes would provide the basis for developing interventions to reduce the risk of diabetes in SSA.

In SSA, intake of a predominantly carbohydrate-rich diet which is low in micronutrients is common [7, 13]. For instance, the intake of animal-based foods is low in SSA [14], and this may be associated with a sub-optimal intake of zinc [15, 16]. The intake of fruits and vegetables could help replenish depleted micronutrients, but the mean fruit and vegetable intake of many people in SSA is lower than the recommended amount and may not reduce the negative impacts of a micronutrient-deficient diet [17, 18]. Micronutrients are potential anti-inflammatory nutrients [19], and zinc supplementation to African children at risk for environmental enteropathy has been associated with improved gut health [20–22]. Vitamin A has established benefits in the maintenance of intestinal epithelia [23, 24], and has been found to reduce intestinal inflammation by significantly increasing the abundance in the intestinal lumen of *Lactobacillus sp* [25], which has antiviral effects [26]. A review of alcohol intake studies suggested that the intake of large amounts of alcohol and its metabolites may affect intestinal epithelium, alter intestinal immune homeostasis, and promote intestinal inflammation through multiple pathways including altering intestinal microbiota composition and function [27]. Thus, a diet deficient in micronutrients or high in alcohol could increase the risk of intestinal inflammation leading to increased intestinal permeability and poor nutrient absorption [28].

Intestinal inflammation is associated with increased intestinal permeability and may lead to translocation of gram-negative bacterial products to the systemic circulation [29–31]. Plasma lipopolysaccharide binding protein (LBP) is a proxy of microbial translocation and is associated with higher C-reactive protein (CRP), a marker of systemic inflammation [32, 33]. Inflammation is associated with activation of immune cells in organs such as the liver, skeletal muscles, adipose tissues, and hypothalamus, resulting in reduced insulin sensitivity and increased risk of diabetes [34].

We hypothesized that a diet which is energy-dense but low in micronutrients or high in alcohol could lead to low-grade intestinal inflammation which could eventually negatively

affect glucose metabolism [35–37]. People living with HIV often experience chronic immune activation and inflammation, even when on antiretroviral therapy (ART), and inflammation from unhealthy diets might affect HIV-infected more than the HIV-uninfected individuals. With this hypothesis, it is essential to explore these associations in populations at varying levels of risk for metabolic disorders. However, the extent to which dietary patterns influence inflammation across these groups is still not well understood. By investigating the associations between diet and inflammation in a mixed population of HIV-infected and uninfected individuals, this study aimed to provide insights into how nutritional interventions may mitigate inflammation and reduce the risk of chronic diseases in HIV-infected and uninfected populations.

## Methods

### Study design and population

This was a cross-sectional analysis of sub-sampled participants of the Role of Environmental Enteropathy on HIV-Associated Diabetes (REEHAD) study, a cross-sectional study investigating the links between environmental enteric dysfunction and diabetes in Mwanza, northwestern Tanzania. Participants were enrolled from 01/05/2019 to 01/05/2020 from both urban and rural area of Mwanza. In this sub-study we included participants with both dietary and inflammation data.

### Sample size estimation

The sample size was calculated using the Open-Source Epidemiologic Statistics for Public Health (OpenEpi) sample size calculator for cross-sectional studies 2013 [38]. In this study, participants had scores for healthy and unhealthy diet patterns, with the diet patterns scores divided into terciles. With assumptions that the proportion of inflammation (intestinal or systemic) in people adhering to healthy and unhealthy diets is 10% and 20% respectively, we needed 600 participants to have an 80% power at 5% significance to detect a 10% difference in proportions in outcomes, and 96% power was met by the available number of samples.

### Participants' characteristics

Data on socio-demographic characteristics including age, sex, employment status, marital status, and education level were collected using a pre-tested structured questionnaire. Data on the possession of assets were collected using a structured questionnaire and used to compute the socio-economic status of the study participants by principal component analysis (PCA) as described elsewhere [10]. Non-communicable diseases behavioral risk factors data, including physical inactivity, was collected using World Health Organization (WHO) Global Physical Activity Questionnaire (GPAQ), and physical activity was computed as metabolic equivalents of tasks (MET) in minutes per week [39]. Smoking status was elicited and grouped as never smoked, past smoker (quit smoking for >1 year) and current smoker (smoking within the past 1 year). Alcohol consumption was grouped as never consumed, past consumption (quit intake for >1 year) and current consumption (consuming within the past 1 year. Details of the classifications have been published in our previous work [10]. Antiretroviral therapy (ART) history was collected from HIV-infected participants' ART cards and verified with ART clinic records.

### Main exposure variables- dietary patterns

A food frequency questionnaire (FFQ) was used to assess dietary habits. Participants were asked to recall the usual intake of food items in terms of frequency and quantity for the past 12

months, and these were then aggregated into 30 food groups based on their nutrient profile and culinary use [40] Table in S1 Table. Dietary patterns were derived using two complementary methods: PCA and reduced rank regression (RRR) as described by Hoffman *et al.* [6]. PCA was used to identify the main dietary patterns based on the variance in dietary data, while RRR was employed to examine how these patterns relate to markers of diabetes, which is hypothesized to link with gut inflammation [7]. The response variables for RRR were selected based on their biological relevance to diabetes, which included waist circumference and BMI. Two patterns were identified from PCA: a vegetable-rich pattern highly loaded with vegetables, fruits, natural fruit juices, bananas, and potatoes; a vegetable-poor pattern highly loaded with artificially sweetened beverages, red meat, alcohol, milk, chips, and crisps; and one pattern was identified by RRR, a carbohydrate-dense pattern highly loaded with grains Table in S2 Table. Details of the diet pattern analyses and results have been described elsewhere [7]. Dietary pattern scores were divided into terciles to simplify the interpretation of associations between diet and inflammation markers. With higher tercile indicating higher adherence to a specific pattern.

## Outcome variables -markers of inflammation

To address our hypothesis that diet may be associated with intestinal inflammation, translocation of microbial products, and systemic inflammation, we chose fecal myeloperoxidase (MPO) and fecal neopterin (NEO) as markers of intestinal inflammation, and plasma levels of LBP and CRP as markers of systemic inflammation. Plasma samples were collected from participants who had fasted ≥8 hours overnight and were aliquoted following standard operating procedures. Stool samples were collected at clinic visits. Plasma and stool aliquots were stored at -80°C until analysis of inflammatory markers at the National Institute for Medical Research Laboratory in Mwanza. Stool samples were analyzed for MPO and NEO while plasma samples were analyzed for LBP and CRP. Biomarkers were analyzed using commercial Human ELISA kits according to manufacturer's instructions. The kits used were from Epitope Diagnostic Inc (San Diego, USA) for MPO, Demeditec Diagnostic (Kiel, Germany) for NEO, and R&D Systems, Bio-Techne brand (Northeast Minneapolis, USA) for LBP. CRP was analyzed in Rigshospitalet, Copenhagen using COBAS-Roche (Basel, Switzerland) [41].

## Anthropometric measurements

Body weight was determined to the nearest 0.1 kg using a digital scale (Seca, Germany) while participants were barefoot and with minimal clothing. Height was measured to the nearest 0.1 cm using a stadiometer fixed to the clinic wall (Seca, Germany). All measurements were taken in triplicate and median values were used for analysis. BMI was categorized as underweight (BMI<18.5kg/m²), normal weight (BMI 18.5-<25 kg/m²), or overweight/obese (BMI≥25 kg/m²).

## Analysis

Data were entered into CSPro and analyzed with STATA 15 (StataCorp, College Station Texas, USA). Background characteristics of the study participants were categorized, and presented as counts (percentages). Markers of inflammation (MPO, NEO, LBP, and CRP) across dietary patterns are presented using box plots. Interaction between markers of inflammation with HIV infection and sex was tested. Pair-wise correlations were used to test for correlations between markers of inflammation. Biomarkers of inflammation were markedly skewed, so two analytical approaches were employed. First, Cuzick's non-parametric test for trend ('nptrend' command in Stata) was deployed to assess the trend of biomarkers across the terciles of dietary

pattern scores. Second, biomarkers of inflammation were divided into quintiles, with higher quintiles representing higher concentrations. We used ordinal logistic regression analysis to assess the associations between dietary pattern terciles and quintiles of the markers of inflammation, controlling for age, sex, and HIV status. We did not adjust for overweight or diabetes in our analysis because we hypothesized that inflammation could lead to diabetes, and overweight is in the causal pathway between diet, inflammation, and diabetes. Adjusting for these factors could have obscured potential causal relationship. Results are presented as odds ratios with 95% confidence intervals. $P$ values<0.05wereconsidered statistically significant.

## Ethical considerations

Ethical clearance for this study was granted by the Medical Research Coordinating Committee of the National Institute for Medical Research (NIMR/HQ/R.8a/Vol IX/2973), and the joint Research Ethics and Review Committee of the Catholic University of Health and Allied Sciences and Bugando Medical Centre (CREC/542/2022). The study strictly adhered to the principles outlined in the Declaration of Helsinki. Participation in the study was contingent upon obtaining written informed consent from all participants before enrollment.

## Results

Of 1173 REEHAD participants, 574 (39%) had diet data, the main exposure variable, and were included in the analysis. There were no differences between REEHAD participants included and those not included in the analysis; the details have been published in our previous work [7]. Of those with diet data, 421 (73%) had LBP data, 424 (74%) had MPO data, 465 (81%) had CRP data and 108 (19%) had NEO data (Fig 1). Just over half were females 339 (59%), 363 (63%) were HIV-infected, and a similar proportion had normal BMI. Of all participants, 324 (57%) were married, 388 (68%) had completed primary level of education, and452 (79%) were self-employed. Only 138 (24%) were current alcohol drinkers, 35 (6%) were current smokers and 26 (4%) were physically inactive (Table 1). No significant interactions between HIV status and the association between dietary patterns and inflammation were observed, and so stratification by HIV status was not done.

To check for consistency between analytical approaches we used both nptrend and ordinal logistic regression, and they both gave similar findings. In ordinal logistic regression analysis

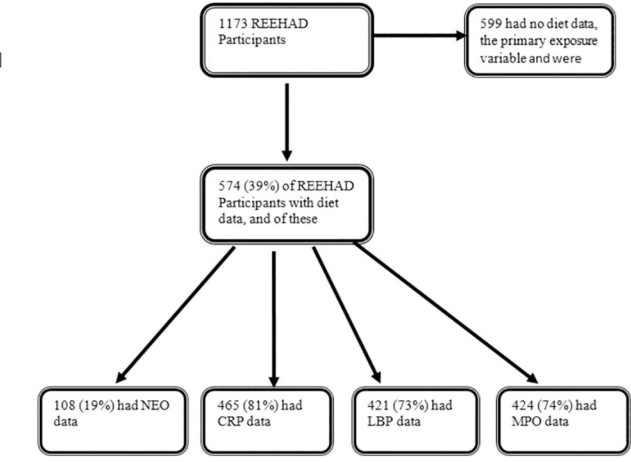

**Fig 1. Participants flow chart.**

**Table 1. Characteristics of the study participants.** N = 574.

| Characteristics | Categories | N (%) |
|---|---|---|
| Age (years) | 18–30 | 95 (16.5) |
| | 31–40 | 178 (31.0) |
| | 41–50 | 168 (29.3) |
| | >50 | 133 (23.2) |
| Sex | Female | 339 (59.1) |
| Education | Never attended school | 107 (18.7) |
| | Primary school | 388 (67.7) |
| | Secondary School and Higher | 78 (13.6) |
| Marital status | Married/cohabiting | 324 (56.6) |
| | Widowed | 86 (15.0) |
| | Separated/divorced | 140(24.4) |
| | Single/never married | 23 (4.0) |
| Employment | Salaried employee | 60 (10.5) |
| | Self-employed | 452 (78.9) |
| | Unemployed/Housewife | 61 (10.6) |
| Smoking | Never | 434 (75.7) |
| | Past-smoker | 104 (18.1) |
| | Current smoker | 35 (6.1) |
| Alcohol | Never | 174 (30.4) |
| | Past-drinker | 261 (45.5) |
| | Current-drinker | 138 (24.1) |
| Physical activity | Active (>= 600 MET minutes/week) | 548 (95.5) |
| | Not Active (<600 MET minutes/week) | 26 (4.5) |
| BMI categories | Underweight (<18.5 kg/m$^2$) | 87 (15.2) |
| | Normal weight (18.5–25.0 kg/m$^2$) | 359 (62.6) |
| | Overweight/obese (>25.0 kg/m$^2$) | 127 (22.2) |
| HIV status | Positive | 363 (63.2) |

BMI, body mass index; MET, Metabolic equivalent of tasks

of the dietary patterns terciles and quintiles of the inflammation markers, we found after that adjusting for age, sex, HIV status and socioeconomic status, vegetable-poor pattern was significantly associated with higher MPO (adjusted OR 1.7, 95% CI 1.1, 2.8). Both vegetable-rich (adjusted OR 2.5, 95% CI1.0, 6.0) and vegetable-poor (adjusted OR 2.4, 95% CI0.9, 6.0) patterns were associated with higher quintiles of NEO. The middle tercile of the carbohydrate-rich pattern was associated with higher quintiles of NEO (adjusted OR 3.6, 95% CI 1.5, 9.1). Table 2. No significant associations were observed between dietary pattern terciles and quintiles of systemic inflammation markers.

Fecal MPO concentrations were higher in those with higher adherence to the vegetable-poor pattern (P-trend 0.03) but were not associated with higher adherence to the vegetable-rich (P-trend 0.4) or carbohydrate-dense patterns (P-trend 0.7). NEO levels were higher in people adhering to the vegetable-rich pattern (P-trend 0.03) with little association with the vegetable-poor (P-trend 0.07) or carbohydrate-dense patterns (P-trend 0.2). LBP and, CRP levels showed no differences across the dietary patterns Fig 2. There were weak correlations between inflammatory markers Table in S3 Table.

We further explored the associations of demographic and lifestyle factors—alcohol drinking, age, SES, HIV, and BMI with fecal and blood markers of inflammation. Table 3 Increasing

**Table 2. Ordinal logistic regression analysis of dietary pattern terciles and inflammatory markers.**

|  | Terciles | Adjusted Odds Ratio (95% CI) | P |
|---|---|---|---|
| **Fecal myeloperoxidase** |  |  |  |
| Vegetable-rich pattern | Lower | ref |  |
|  | Middle | 1.1 (0.7, 1.6) | 1.00 |
|  | Upper | 1.2 (0.9, 1.8) | 0.40 |
| Vegetable-poor pattern | Lower | ref |  |
|  | Middle | 1.5 (1.0, 2.3) | 0.06 |
|  | Upper | **1.9 (1.2, 2.9)** | **0.01** |
| Carbohydrate-dense pattern | Lower | ref |  |
|  | Middle | 1.0 (0.6, 1.5) | 1.0 |
|  | Upper | 1.2 (0.8, 1.8) | 0.4 |
| **Fecal neopterin** |  |  |  |
| Vegetable-rich pattern | Lower | ref |  |
|  | Middle | 1.8 (0.8, 4.3) | 0.20 |
|  | Upper | **2.7 (1.1, 6.5)** | **0.02** |
| Vegetable-poor pattern | Lower | ref |  |
|  | Middle | **3.4 (1.4, 8.2)** | **0.01** |
|  | Upper | **2.6 (1.0, 6.4)** | **0.048** |
| Carbohydrate-dense pattern | Lower | ref |  |
|  | Middle | **3.6 (1.4, 8.9)** | **0.01** |
|  | Upper | 2.1 (0.9, 5.2) | 0.10 |
| **Plasma C-Reactive Protein** |  |  |  |
| Vegetable rich pattern | Lower | ref |  |
|  | Middle | 0.9 (0.6, 1.3) | 0.60 |
|  | Upper | 0.9 (0.6, 1.4) | 0.70 |
| Vegetable poor pattern | Lower | ref |  |
|  | Middle | 1.1 (0.7, 1.6) | 0.58 |
|  | Upper | 1.0 (0.6, 1.5) | 0.98 |
| Carbohydrate dense pattern | Lower | ref |  |
|  | Middle | 1.1 (0.7, 1.7) | 0.62 |
|  | Upper | 1.4 (0.9, 2.2) | 0.08 |
| **Plasma Lipopolysaccharide Binding Protein** |  |  |  |
| Vegetable rich pattern | Lower | ref |  |
|  | Middle | 1.2 (0.8, 1.9) | 0.35 |
|  | Upper | 1.2 (0.8, 1.9) | 0.31 |
| Vegetable poor pattern | Lower | ref |  |
|  | Middle | 1.1 (0.7, 1.6) | 0.71 |
|  | Upper | 1.1 (0.7, 1.7) | 0.69 |
| Carbohydrate dense pattern | Lower | ref |  |
|  | Middle | 1.1 (0.8, 1.8) | 0.51 |
|  | Upper | 1.2 (0.8, 1.8) | 0.51 |

All Models were controlled for age, sex, and HIV status

in age was associated with increases in all markers except neopterin. MPO was associated with higher alcohol intake (adjusted OR 1.3, 95% CI 1.0, 1.6). CRP was associated with overweight/ obesity (adjusted OR 2.3, 95% CI 1.3, 4.1) and HIV positivity (adjusted OR 2.1 95% CI 1.5, 3.0. LBP was associated with female sex (adjusted OR 1.6, 95% CI 1.1, 2.3.

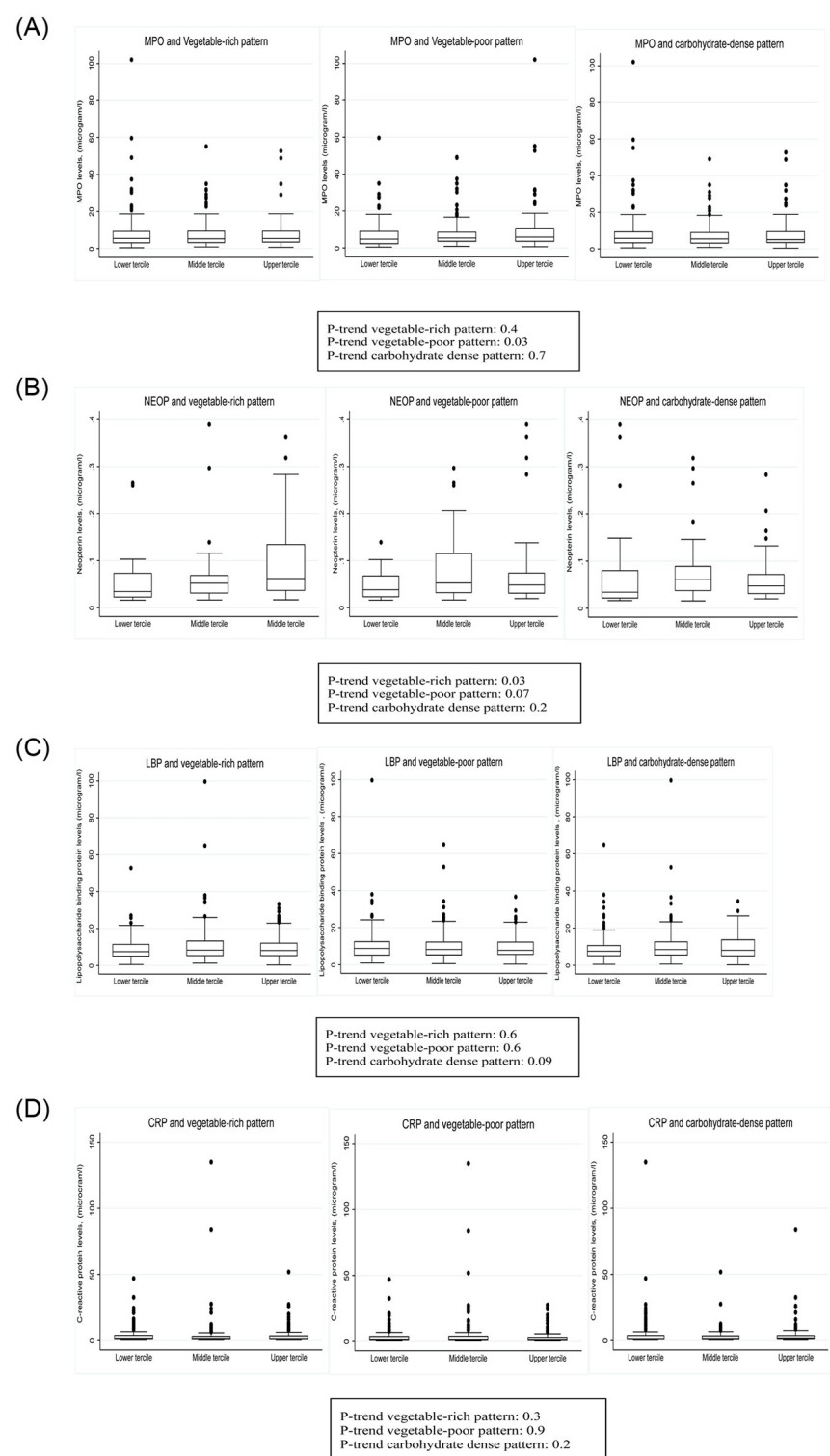

**Fig 2. A**: Distribution of fecal myeloperoxidase (MPO) across the terciles of vegetable-rich, vegetable-poor and carbohydrate-dense dietary patterns. **B**: Distribution of fecal neopterin across the terciles of vegetable-rich, vegetable-poor and carbohydrate-dense dietary patterns. **C**: Distribution of plasma lipopolysaccharide binding protein (LBP) across the terciles of vegetable-rich, vegetable-poor and carbohydrate-dense dietary patterns. **D**: Distribution of plasma C—reactive protein (CRP) across the terciles of vegetable-rich, vegetable-poor and carbohydrate-dense dietary patterns.

**Table 3. Ordinal logistic regression analysis: Associations between markers of inflammation (outcome) and participants' characteristics.**

| | | Fecal Myeloperoxidase | | Fecal Neopterin | | Plasma C-reactive protein | | Plasma Lipopolysaccharide binding protein | |
|---|---|---|---|---|---|---|---|---|---|
| | | OR (95% CI) | P | OR (95% CI) | P | OR (95% CI) | P | OR (95% CI) | P |
| Sex | Female | 0.7 (0.5, 1.0) | 0.1 | 1.0 (0.5, 2.2) | 0.9 | 1.1 (0.7, 1.5) | 0.8 | 1.6 (1.1, 2.3) | **0.01** |
| Age Categories | 18–30 | Reference | | Reference | | Reference | | Reference | |
| | 31–40 | 1.9 (1.1, 3.3) | **0.01** | 0.6 (0.2, 1.8) | 0.4 | 1.1 (0.7, 1.3) | 0.8 | 1.2 (0.7, 2.0) | 0.5 |
| | 41–50 | 2.2 (1.3, 3.8) | **0.003** | 0.8 (0.3, 2.4) | 0.8 | 2.1 (1.2, 3.4) | **0.004** | 1.3 (0.8, 1.4) | 0.3 |
| | >50 | 3.2 (1.8, 5.6) | **0.001** | 0.5 (1.4, 1.6) | 0.2 | 3.4 (2.0, 5.7) | **0.001** | 1.8 (1.0, 3.1) | **0.04** |
| BMI categories | Underweight | Reference | | Reference | | Reference | | Reference | |
| | Normal weight | 0.9 (0.6, 1.5) | 0.8 | 1.8 (0.5, 6.6) | 0.4 | 1.1 (0.7,1.8) | 0.6 | 0.9 (0.6, 1.5) | 0.8 |
| | Overweight/Obesity | 1.4 (0.8) | 0.3 | 2.3 (0.5, 10.3) | 0.3 | 2.3 (1.3, 4.1) | **0.01** | 0.9 (0.5, 1.6) | 0.6 |
| Socio-economic status | Low | Reference | | Reference | | Reference | | Reference | |
| | Middle | 1.0 (0.6, 1.5) | 1.0 | 1.2 (0.5, 2.9) | 0.7 | 1.2 (0.8, 1.7) | 0.4 | 1.1 (0.7, 1.7) | 0.7 |
| | High | 1.3 (0.9, 2.1) | 0.9 | 1.8 (0.8, 4.3) | 0.1 | 1.4 (0.9, 2.1) | 0.1 | 1.1 (0.7, 1.7) | 0.8 |
| Alcohol intake | No | Reference | | Reference | | Reference | | Reference | |
| | Yes | 1.3 (1.0, 1.6) | **0.04** | 1.1 (0.7, 1.8) | 0.7 | 1.1 (0.8, 1.3) | 0.6 | 1.1 (0.8, 1.4) | 0.5 |
| HIV | Negative | Reference | | Reference | | Reference | | Reference | |
| | Positive | 1.2 (0.8, 1.7) | 0.3 | 1.6 (0.8, 3.2) | 0.2 | 2.1 (1.5, 3.0) | **0.001** | 1.0 (0.7, 1.4) | 0.9 |

All markers of inflammation were divided into quintiles, while alcohol taking was categorized as yes if a current drinker and no if participant had never drank or had quit for more than a year. All models have been adjusted for sex, age, socioeconomic status and HIV status.

## Discussion

Our study found that vegetable-poor diets were associated with increased levels of intestinal inflammation markers, particularly MPO, and NEO. These associations were not observed with systemic inflammation markers such as LBP and CRP. This suggests that while diet influences gut inflammation, it may not directly translate to systemic inflammation in the studied population. The weak correlation observed between these biomarkers suggests that they may reflect different domains of enteropathy, highlighting the complexity of using single biomarkers to assess gut inflammation comprehensively [42].

The vegetable-poor pattern was high in alcohol, red meat, artificially sweetened beverages, milk, chips, and crisps [7]. Alcohol and spicy foods have pro-inflammatory potential and are thought to increase levels of MPO as they trigger intestinal inflammatory response [43]. We observed high levels of fecal MPO in participants who reported to be current alcohol drinkers. This may be explained by alcohol-induced changes in the gut microbiota composition and metabolic function contributing to alcohol-induced oxidative stress, and intestinal inflammation [44]. NEO levels have also been associated with high alcohol intake as was seen in alcoholic cirrhotic patients [45], although it was not associated with alcohol drinking in the current study. However, the association of alcohol with intestinal inflammation is not well understood and needs to be further studied [46].

Moderate intake of the carbohydrate-dense pattern was associated with increased odds of high NEO levels; although the confidence interval for the adjusted OR was wide and included 1, high intake of this pattern also showed a trend to higher NEO. This discrepancy could be due to statistical fluctuations or a smaller sample size. A possible explanation is that a carbohydrate-rich diet negatively affects the microbiome diversity [37]. Our study showed increased odds of high NEO in participants with a high intake of a vegetable-rich pattern which was

unexpected and in contrast to a Iranian study that showed a significant lowering of plasma NEO following intake of spinach extract [47]. These results are hard to explain and could be chance, in part because the sample size for NEO was low, but could also be because we analyzed fecal and not plasma neopterin as the study in Iran. We observed previously that in our cohort, the vegetable-rich pattern was not protective against diabetes, and was associated with an increased risk of prediabetes, contrary to other literature [7] but probably explained by increased in intestinal inflammation reported in the current analysis. The preparation of these vegetables by our participants may involve pro-inflammatory foods such as saturated fats; this needs further studies.

We found no association between CRP and adherence to any of the three patterns of diet. Our findings are in line with a prospective study which found no significant CRP changes between those adhering to healthy patterns and those adhering to unhealthy dietary patterns [48]. However, the data suggested that dietary factors may act as independent risk factors mediated through BMI [49]; this agrees with our study, where plasma CRP levels were significantly higher in participants with higher BMI. Lower levels of CRP were also observed in participants adhering to a healthy diet rich in dietary fibers such as whole grains [50] suggesting that a higher intake of whole grains may reduce the risk of systemic inflammation [36, 51].

This is among the few studies that have explored population dietary patterns and their association with markers of intestinal and systemic inflammation. Virtually none of the participants knew they had diabetes at the time of recruitment, reducing the likelihood that diabetes management influenced their dietary intake and inflammatory status. Despite the strengths of our study, several limitations should be acknowledged. First, this was a cross-sectional study and causality cannot be confirmed. Our analysis was based on dietary patterns derived from a food frequency questionnaire (FFQ), which is subject to recall bias and could lead to misclassification of dietary intake. Also, for some outcomes, particularly NEO, the sample size was small and this could explain the inconsistent results. Additionally, we did not have data on the gut microbiota or other intestinal microbes, such as worms and protozoa, which are known to influence gut inflammation. The absence of this data means we could not directly assess the interaction between dietary patterns and these microbial communities. Future research should include microbiome analyses to provide a more comprehensive understanding of the mechanisms linking diet to intestinal inflammation.

In conclusion, our study found that vegetable-poor diets are associated with intestinal inflammation but not systemic inflammation in Tanzanian adults. These findings suggest that the impact of diet on non-communicable diseases might be mediated through pathways which do not include the marker of systemic inflammation we used in this study. The unexpected association between the vegetable-rich pattern and neopterin highlights the need for further research into the preparation methods and overall dietary context in Tanzania. Future research should explore these alternative pathways and include microbiome analyses to provide a more comprehensive understanding of the relationship between diet, gut health, and non-communicable diseases.

## Supporting information

**S1 Table. Food groupings used in dietary patterns analyses.**
(DOCX)

**S2 Table. Factor loadings of the factors retained by principal component analysis and reduced rank regression analysis-derived dietary patterns.**
(DOCX)

**S3 Table. Pair-wise correlation of the markers of inflammation.**
(DOCX)

## Acknowledgments

The authors extend their gratitude to all study participants. Special acknowledgment is given to the late Jonas Anosisye Aswile for his invaluable contribution in handling diet data. Heartfelt appreciation is also expressed to the National Institute for Medical Research and the Catholic University of Health and Allied Sciences for their unwavering support during this research.

## Author Contributions

**Conceptualization:** Evangelista Kenan Malindisa, Haruna Dika, Mette Frahm Olsen, Rikke Krogh-Madsen, Henrik Friis, Daniel Faurholt-Jepsen, Paul Kelly, Suzanne Filteau, George PrayGod.

**Data curation:** Evangelista Kenan Malindisa, Andrea Mary Rehman, Belinda Kweka, Ruth Frikke-Schmidt, Paul Kelly, George PrayGod.

**Formal analysis:** Evangelista Kenan Malindisa, Andrea Mary Rehman, Belinda Kweka, Jim Todd, Paul Kelly, Suzanne Filteau, George PrayGod.

**Funding acquisition:** Henrik Friis, Daniel Faurholt-Jepsen, Suzanne Filteau, George PrayGod.

**Investigation:** Evangelista Kenan Malindisa, Paul Kelly.

**Methodology:** Evangelista Kenan Malindisa, Haruna Dika, Belinda Kweka, Mette Frahm Olsen, Rikke Krogh-Madsen, Henrik Friis, Daniel Faurholt-Jepsen, Paul Kelly, Suzanne Filteau, George PrayGod.

**Supervision:** Haruna Dika, Henrik Friis, Daniel Faurholt-Jepsen, Suzanne Filteau, George PrayGod.

**Visualization:** Evangelista Kenan Malindisa.

**Writing – original draft:** Evangelista Kenan Malindisa.

**Writing – review & editing:** Evangelista Kenan Malindisa, Haruna Dika, Andrea Mary Rehman, Belinda Kweka, Jim Todd, Mette Frahm Olsen, Rikke Krogh-Madsen, Ruth Frikke-Schmidt, Henrik Friis, Daniel Faurholt-Jepsen, Paul Kelly, Suzanne Filteau, George PrayGod.

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
