## [Decision Letter · Decision Letter 0]

8 Sep 2024

PONE-D-24-26062Associations between dietary patterns and intestinal inflammation among HIV-infected and uninfected adults: a cross-sectional study in TanzaniaPLOS ONE

Dear Dr. Malindisa,

Thank you for submitting your manuscript to PLOS ONE. After careful consideration, we feel that it has merit but does not fully meet PLOS ONE’s publication criteria as it currently stands. Therefore, we invite you to submit a revised version of the manuscript that addresses the points raised during the review process.

We look forward to receiving your revised manuscript.

Kind regards,

Sepiso K. Masenga, PhD

Academic Editor

PLOS ONE

Journal Requirements:

"This project is part of the EDCTP2 program supported by the European Union (grant agreement number: TMA2017GSF-1965-REEHAD), awarded to GP. It received additional support from the Ministry of Foreign Affairs of Denmark through the DANIDA Fellowship Centre (grant: 16-P01-TAN) awarded to GP. The funders are not responsible for any use that may be made of the information contained in this publication"

3. Please expand the acronym “EDCTP” (as indicated in your financial disclosure) so that it states the name of your funders in full.

Reviewers' comments:

Reviewer's Responses to Questions

**Comments to the Author**

1. Is the manuscript technically sound, and do the data support the conclusions?

Reviewer #1: Partly

Reviewer #2: Yes

2. Has the statistical analysis been performed appropriately and rigorously? 

Reviewer #1: Yes

Reviewer #2: Yes

3. Have the authors made all data underlying the findings in their manuscript fully available?

Reviewer #1: Yes

Reviewer #2: Yes

4. Is the manuscript presented in an intelligible fashion and written in standard English?

Reviewer #1: Yes

Reviewer #2: Yes

5. Review Comments to the Author

Reviewer #1: This manuscript explores a relevant topic of the link between diet and inflammatory markers. What is especially important is that it analyses diet as dietary patterns instead of individual nutrients and food groups which better reflects actual eating patterns and this is commendable. My biggest issue or point for clarification is the fact the authors make a distinction between HIV infected and uninfected individuals in the title, however this then not addressed in the manuscript. I would appreciate if the authors could clarify.

Introduction:

This section would benefit from a better explained rationale that is followed by the aims of the study. This can then be followed by a hypothesis but the aim should not be omitted.

Also, the title clearly makes a distinction between HIV infected and uninfected individuals but this is not addressed in the introduction. There should also be a rationale provided for this.

Methods and results:

I do not see any distinctions or analyses made to explore if there would be differences in results between HIV+ and HIV- participants, apart from Table 3. If this is indeed one of the objectives, then it should be addressed. This should then also be highlighted in discussion.

Reviewer #2: The present study titled “Associations between dietary patterns and intestinal inflammation among HIV-infected and uninfected adults: a cross-sectional study in Tanzania” has been conducted. Inflammatory diseases has unfortunately become a common disease and nutritional research in this field is my interest.

The authors have written well to present their findings.

- Please provide a table of how many food groups were included in the PCA and RRR ? And what were the components of each group?

- Please provide a table of factor loading of food groups in each pattern, like the reference below:

https://link.springer.com/article/10.1186/s12889-022-13166-0

6. PLOS authors have the option to publish the peer review history of their article (what does this mean?). If published, this will include your full peer review and any attached files.

Reviewer #1: **Yes: **Leta Pilic

Reviewer #2: No

---

## [Author Response · Author response to Decision Letter 0]

16 Sep 2024

Editor

 1. Please ensure that your manuscript meets PLOS ONE's style requirements, including those for file naming

Response: Thank you, we have done that as advised

2.Please state what role the funders took in the study

Response:Thank you, we have stated in the cover letter that, the funders had no role in study design, data collection and analysis, decision to publish, or preparation of the manuscript

3. Please expand the acronym “EDCTP”

Response: Thank you, we have now expanded as advised

4. Please amend your list of authors on the manuscript to ensure that each author is linked to an affiliation

Response: Thank you,

This was confirmed

5. Please include captions for your Supporting Information files at the end of your manuscript, and update any in-text citations to match accordingly

Response:Thank you,

This has been done accordingly

6. Please review your reference list to ensure that it is complete and correct

Response: Thank you,

We have confirmed

Reviewer 1

1. This manuscript explores a relevant topic of the link between diet and inflammatory markers. What is especially important is that it analyses diet as dietary patterns instead of individual nutrients and food groups which better reflects actual eating patterns and this is commendable

Response: We appreciate your dedication to review and contribution to the improvement of this manuscript

2. My biggest issue or point for clarification is the fact the authors make a distinction between HIV infected and uninfected individuals in the title, however this then not addressed in the manuscript. I would appreciate if the authors could clarify.

Response: Thank you for highlighting this important point. We acknowledge the concern regarding the mention of HIV-infected and uninfected individuals in the title, while this distinction was not addressed in the manuscript. Initially, our study aimed to explore whether HIV status modifies the association between dietary patterns and inflammation, as inflammation is a key factor in both HIV progression and metabolic dysfunction.

Upon performing the analyses, we found no significant interaction between HIV status and the association between dietary patterns and inflammation. Consequently, we chose not to stratify the results by HIV status and instead adjusted for HIV as a confounding variable in the models

However, this cross-sectional study was nested in a cohort with HIV-positive and negative participants, 

We thought having HIV in the title could best characterize our study participants.

We have added a text in results section to explain this

2. Introduction:

This section would benefit from a better explained rationale that is followed by the aims of the study. This can then be followed by a hypothesis but the aim should not be omitted

Response: We have revised the introduction to address this. 

3. Also, the title clearly makes a distinction between HIV infected and uninfected individuals but this is not addressed in the introduction. There should also be a rationale provided for this.

Response: We have responded this in the first comment, and the rationale have been provided.

4. Methods and results:

I do not see any distinctions or analyses made to explore if there would be differences in results between HIV+ and HIV- participants, apart from Table 3. If this is indeed one of the objectives, then it should be addressed. This should then also be highlighted in discussion

Response: We have responded this in the first response

Reviewer 2

1. The present study titled “Associations between dietary patterns and intestinal inflammation among HIV-infected and uninfected adults: a cross-sectional study in Tanzania” has been conducted. Inflammatory diseases has unfortunately become a common disease and nutritional research in this field is my interest.

The authors have written well to present their findings.

Response: Thank you for this comment and for reviewing our manuscript.

2. Please provide a table of how many food groups were included in the PCA and RRR? And what were the components of each group?

Response: We have added this in a supplementary table. 

3. Please provide a table of factor loading of food groups in each pattern, like the reference below:

https://link.springer.com/article/10.1186/s12889-022-13166-0

Response: We have added this in a supplementary table

---

## [Editor Report · Decision Letter 1]

24 Sep 2024

Associations between dietary patterns and intestinal inflammation among HIV-infected and uninfected adults: a cross-sectional study in Tanzania

PONE-D-24-26062R1

Dear Dr. Malindisa,

We’re pleased to inform you that your manuscript has been judged scientifically suitable for publication and will be formally accepted for publication once it meets all outstanding technical requirements.

Kind regards,

Sepiso K. Masenga, PhD

Academic Editor

PLOS ONE
---

## [Editor Report · Acceptance letter]

8 Oct 2024

PONE-D-24-26062R1 

PLOS ONE

Dear Dr. Malindisa, 

I'm pleased to inform you that your manuscript has been deemed suitable for publication in PLOS ONE. Congratulations! Your manuscript is now being handed over to our production team.

Kind regards, 

on behalf of

Prof. Sepiso K. Masenga 

Academic Editor

PLOS ONE